# Star-Agents: Automatic Data Optimization with LLM Agents for Instruction Tuning

**Hang Zhou**[1,2], **Yehui Tang**[2], **Haochen Qin**[2], **Yujie Yang**[2], **Renren Jin**[1],
**Deyi Xiong**[1*], **Kai Han**[2*], **Yunhe Wang**[2*]
[1]College of Intelligence and Computing, Tianjin University, Tianjin, China.
[2]Huawei Noah's Ark Lab.
{zhouhang25, yehui.tang, qinhaochen1, yangyujie26}@huawei.com,
{rrjin, dyxiong}@tju.edu.cn, {kai.han, yunhe.wang}@huawei.com.

## Abstract

The efficacy of large language models (LLMs) on downstream tasks usually hinges on instruction tuning, which relies critically on the quality of training data. Unfortunately, collecting high-quality and diverse data is both expensive and time-consuming. To mitigate this issue, we propose a novel Star-Agents framework, which automates the enhancement of data quality across datasets through multi-agent collaboration and assessment. The framework adopts a three-pronged strategy. It initially generates diverse instruction data with multiple LLM agents through a bespoke sampling method. Subsequently, the generated data undergo a rigorous evaluation using a dual-model method that assesses both difficulty and quality. Finaly, the above process evolves in a dynamic refinement phase, where more effective LLMs are prioritized, enhancing the overall data quality. Our empirical studies, including instruction tuning experiments with models such as Pythia and LLaMA, demonstrate the effectiveness of the proposed framework. Optimized datasets have achieved substantial improvements, with an average increase of 12% and notable gains in specific metrics, such as a 40% improvement in Fermi, as evidenced by benchmarks like MT-bench, Vicuna bench, and WizardLM testset. Codes will be released soon[1].

## 1 Introduction

The research and development of natural language understanding and generation have been dramatically accelerated with the emergence and prevalence of LLMs [39, 31, 30]. These models have been extensively applied in a wide range of scenarios, e.g., question answering and text generation, significantly enhancing downstream task performance due to their exceptional ability to follow instructions [3, 53, 49, 10, 28]. Such an instruction-following capability is primarily acquired through a process known as instruction tuning [40, 23, 5], where LLMs are fine-tuned on instruction data. It is hence widely acknowledged that the quality of instructions plays a pivotal role [5, 20, 48, 29].

Historically, the creation of instruction data for training LLMs has heavily relied on the expertise of human annotators, as evidenced by substantial scholarly contributions [14, 50, 41, 38, 9, 27, 21]. While expert-driven data generation assures the production of high-quality instructions, the enormous volume of data necessary for effective training renders this method economically untenable. In response, recent efforts have shifted towards the utilization of LLMs to automatically generate instructions, thereby mitigating the reliance on costly human annotation [37, 32, 44, 18]. Concurrently,

---

[*]Corresponding authors.
[1]https://github.com/CANGLETIAN/Star-Agents

there is a growing emphasis on the generation and selection of challenging examples, grounded in the belief that more complex and difficult instructions can substantially elevate model capabilities [22, 17].

Despite the clear advantages of using LLMs for data generation, several challenges persist in this strategy. Primarily, previous efforts often depend on a single LLM, resulting in data that may lack stylistic variety [4] and encompass a limited range of difficulty levels , which may not be ideal for all models. Additionally, there is a trend towards the creation of exceedingly complex instructions [19, 44, 18], which may surpass the operational capabilities of models with small parameter scale, thereby hindering their ability to fully capitalize on the data's potential for performance enhancement.

To address the aforementioned challenges, we propose the **Star-Agents** framework, an advanced automatic data optimization system specifically designed to learn and refine instruction samples with suitable complexity and diversity for target LLMs. The framework consists of three main components. First, to increase the diversity of generated data, an instruction data rewriting process involving multiple advanced LLM agents is proposed. This process samples different LLM agents for rewriting instructions and responses separately (referred to as agent-pairs). Next, to select high-quality samples, the generated data undergo a dual-model evaluation function with appropriate complexity as the selection metric. Finally, to balance data diversity and quality, the sampling probability of agent-pairs is adjusted and evolved based on the composite scores of the selected data, identifying agent-pairs that generate high-quality data.

Extensive experiments are conducted to evaluate instruction-following capabilities of LLMs on a variety of benchmark datasets, including MT-bench [54], Vicuna-bench [54], and the WizardLM testset [44]. Instruction tuning experiments with LLMs such as Pythia and LLaMA, demonstrate the effectiveness of the Star-Agents framework. LLMs trained on data generated by Star-Agent outperform those (the same LLMs) trained on the Evol-Instruct dataset [44] or data selected according to the Instruction-Following Difficulty (IFD) metric [20]. Significantly, the optimized datasets have resulted in an average performance improvement of 12%, with some metrics such as Fermi demonstrating gains of over 40%.

## 2 Related Work

Our work is related to both instruction data generation and selection. We briefly review these topics within the constraint of space.

**Instruction Data Generation**   Datasets like Dolly [7] and OpenAssistant [15] are built from human-generated instruction data. The ShareGPT dataset, built from conversations between humans and ChatGPT, has been effectively used to improve the instruction-following performance of fine-tuned models [6]. Both Self-Instruct [36] and Alpaca [33] leverage the generation capabilities of GPT-3 to expand seed instructions. The generated instructions undergo filtering to eliminate low-quality instructions while the kept instructions are used to fine-tune the model to enhance the model's ability to respond to instructions. Baize [45] proposes a self-dialogue framework, using questions from popular Q&A websites as starting topics, then having LLMs converse with themselves. CAMEL [16] introduces a role-playing framework where LLMs discuss a given topic when playing a role as either "user" or "assistant". UltraChat [8] uses real-world named entities combined with various text-writing tasks to generate diverse and high-quality multi-turn dialogues for LLMs. Lion [13] introduces the concept of adversarial distillation, using the Imitation-Discrimination-Generation stages to iteratively generate data, refine existing instructions, and produces more complex and diverse instructions to expand the capabilities of the student model. Evol-Instruct [44] uses five manually designed prompts to explicitly guide LLM in rewriting existing simple instructions into more complex ones. The WizardLM model, trained with Evol-Instuct, ranks highly on MT-Bench [54], highlighting the importance of data quality in training effective LLMs.

**Instruction Data Selection**   With the aforementioned methods, it is not difficult to use LLMs to generate large instruction tuning datasets at low cost. However, for instruction-tuned language models, data quality is more crucial than quantity. In this aspect, ALPAGASUS [5] evaluates the effectiveness of instruction data by leveraging ChatGPT. INSTAG [24] automatically generates tags for instruction samples with ChatGPT and keeps diversity by selecting subsets with more tags. Cherry LLM [20] pioneers the self-guided approach, using the IFD metric to measure the difficulty for an LLM to learn

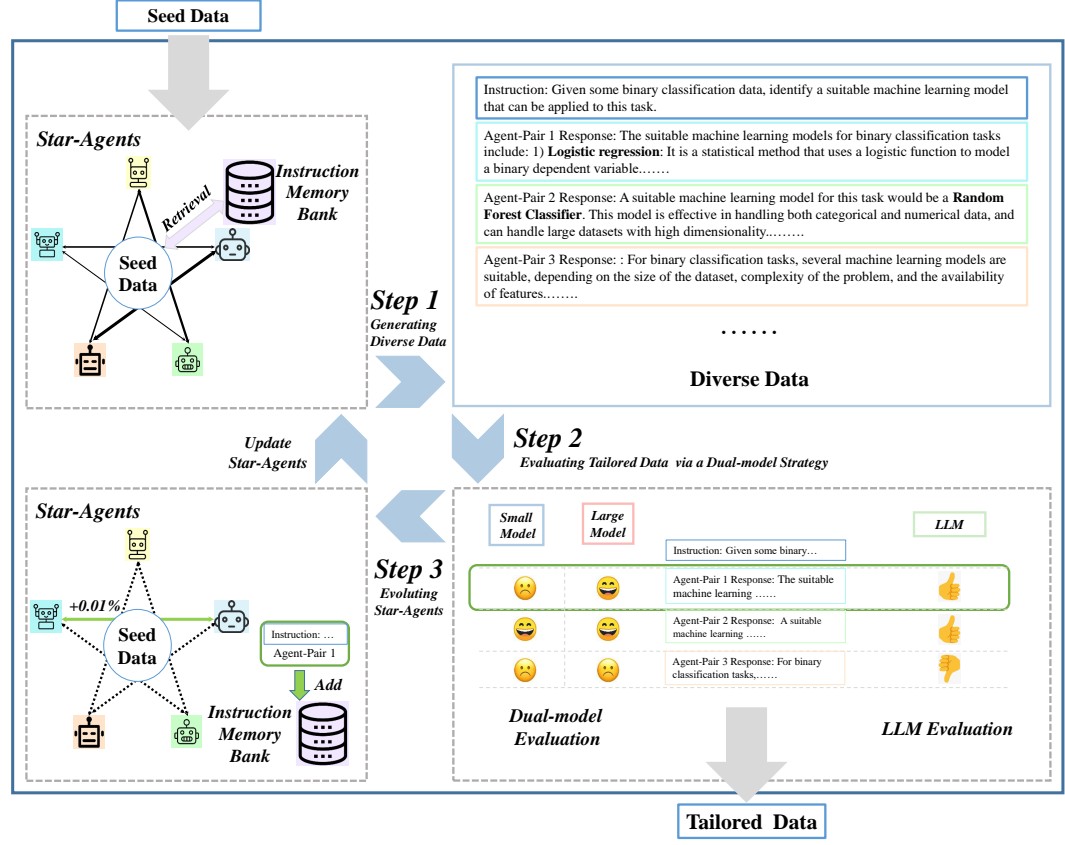

Figure 1: The diagram of the Star-Agents Framework. Step 1 is designed to gather diverse instructions and responses as shown in Appendix A.3. Step 2 focuses on selecting high-quality, tailored data from the data collected in Step 1. Finally, Step 3 aims to enhance the effectiveness and efficiency of the data generation process by evolving the Star-Agents framework.

an instruction sample. This allows to select instruction samples that significantly enhance training efficiency without resorting to an external model. DEITA [22] first uses ChatGPT to evaluate the complexity and quality of samples, then assesses the diversity of samples based on the distance between model embeddings, thereby guaranteeing complexity, quality, and diversity in the subset. LIFT [46] first guides GPT-4 to generate challenging instructions to expand the data distribution and then uses dimensionality reduction and row variance analysis to select representative high-quality data, where GPT-4 generates a quality score for each instruction. LESS [43] first stores the gradient features of samples in the dataset, then calculates the similarity between a small number of samples from the target task and the training data samples. Based on the calculated similarity scores, it selects the training samples whose gradient features are most similar to those of the target task samples as the fine-tuning instances. Data selection not only improves training efficiency but also prevents low-quality or poison data from undermining model performance by filtering them out [47].

## 3  Star-Agents

The aim of our research is to construct a high-quality dataset $T$ of tailored complexity for the target LLM through the enhancement of an initial seed dataset $S = (I_i, R_i)_{i=1}^N$, consisting of instruction-response pairs $(I, R)$.

To this end, we introduce the Star-Agents Framework, depicted in Figure 1, which is segmented into three steps. The first step leverages a spectrum of advanced LLMs, each trained independently. These models are engaged in a dynamic interaction to generate a diverse data candidate set $D(S_i)$ by sampling agent-pair derived from $S_i$ as detailed in Section 3.1. Following this, we apply a dual-

model evaluation strategy $\pi(\cdot)$ to meticulously extract the most suitable data from $D(S_i)$, aiming to substantially elevate the target model's performance. This process is elaborated in Section 3.2. To enhance the effectiveness and efficiency of the Star-Agents framework in generating tailored data, we have developed an evolutionary strategy for the Star-Agents, as discussed in Section 3.3. After these three steps, a tailored high-quality dataset $T$ is obtained from the seed dataset, which is formulated as:

$$T = \{\arg\max_{d \in D(S_i)} \pi\left(D(S_i)\right) \mid i = 1, 2, \cdots, N\}. \tag{1}$$

## 3.1 Generating Diverse Data

To improve the instruction-tuned model, it is crucial to assemble a high-quality and diverse instruction dataset [22]. Traditional methods often use a single LLM, such as ChatGPT, for data enrichment. In contrast, our approach employs multiple LLMs to avoid monotonous data distribution. This multifaceted strategy also addresses the limitations and risks of quality degradation on domain-specific tasks associated with using a single model. To counter these challenges, we propose to use an Agent-Pair strategy.

**Agent-Pair.** Utilizing a spectrum of LLMs, each trained with discrepant setting, facilitates the generation of varied responses to given instructions. This diversity is crucial for synthesizing a dataset characterized by high richness [24].

The Star-Agents framework strategically pairs different LLMs to rewrite the instructions in the seed dataset and generate new responses to increase the diversity. With agent-pair $(A_j^I, A_k^R)$, a new instruction data can be generated as follows:

$$f_{j,k}\left(I_i, R_i\right) = (A_j^I(I_i), A_k^R(R_i)), \tag{2}$$

where $A^I$ and $A^R$ represent the agents that rewrites the instruction and response to the instruction, respectively.

Given the high cost of deploying all agent-pairs, a feasible solution to balance cost and agent diversity is to sample a subset of agent-pairs from the Star-Agents for data generation. Equation 3 formulates this process, where $D$ is collected dataset generated by performing $f$ over all sampled pairs $(A_j^I, A_k^R)$ of instruction agents $A_j^I$ and response agent $A_k^R$ with sampling probabilities $p_{jk}$:

$$D\left(S_i\right) = \{f_{j_1,k_1}\left(S_i\right), \cdots, f_{j_M,k_M}\left(S_i\right) \mid (j_m, k_m) \sim p_{jk}, m = 1, 2, \cdots, M\}, \tag{3}$$

$M$ is number of agent-pairs sampled for a single seed sample. The sampling probability $p_{jk}$ is initialized as a uniform distribution and will be updated using the method described in Subsection 3.3 during data generation. Meanwhile, an Instruction Memory Bank that stores high-quality instructions will be updated. To ensure the lower bound of data quality, each iteration will consistently call a fixed set of agent-pairs, referred to as base agent-pairs.

## 3.2 Evaluating Tailored Data via a Dual-model Strategy

Identifying and selecting tailored data from a diverse dataset is crucial for enhancing model performance, especially since the presence of low-quality data can impede model functionality. It is acknowledged that data samples that are lengthy, complex, and challenging significantly benefit the instruction tuning process [22].

Nevertheless, too complex instruction data may be not necessarily benefit model performance. We have observed that for models with 14M and 70M parameters as illustrated in Figure 2, the Evol-Instruct dataset, though more challenging than the Alpaca dataset, results in diminished model performance. This suggests that intricate examples may surpass the capabilities of small models and be harmful for model performance, despite the advantages of using complex data for large models.

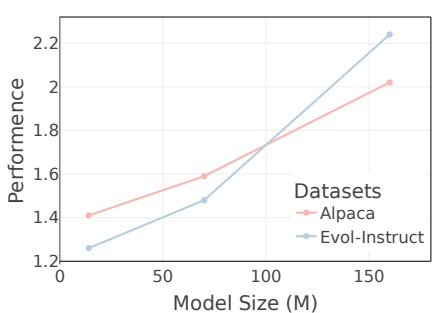

Figure 2: Performance comparison of varied-scale models on the Alpaca and Evol-Instruct datasets. The tasks from the Evol-Instruct dataset are more complex than those from Alpaca.

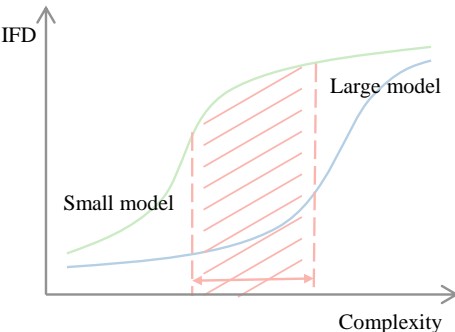

Figure 3: Illustration of dual-model evaluation. Data with a significant gap between the IFD scores of the small and large models will be prioritised.

**Dual-model Evaluation.** To address the issue mentioned above, we propose to use a larger model to evaluate the difficulty of data instances together with the evaluation from a smaller model (target LLM), hence termed as dual-model evaluation. Inspired by Cherry LLM [20], we employ the IFD metric to measure the degree of difficulty a data sample presents to the target model, which is calcuated as

$$\text{IFD}(I_i, R_i) = \frac{\exp\left(-\frac{1}{|R_i|} \sum_{w \in R_i} \log P(w|I_i)\right)}{\exp\left(-\frac{1}{|R_i|} \sum_{w \in R_i} \log P(w)\right)}. \tag{4}$$

We assume that for the same sample, stronger model yields a smaller IFD score. When the IFD scores of the two models are close to each other, it indicates that the sample is either too simple or too complex, which is not contributive to effective learning. However, when their IFD scores differ significantly, it indicates that the data is sufficiently complex for the smaller model but still within the capability range of the stronger model. This is a tailored complexity for facilitating learning. The above data assessment method is illustrated at Figure 3 and formulated as

$$\pi_{\text{dual}}^i = \frac{\text{IFD}_{\text{small}}(I_i, R_i) - \text{IFD}_{\text{large}}(I_i, R_i)}{\max_{1 \le i \le m} \left(\text{IFD}_{\text{small}}(I_i, R_i) - \text{IFD}_{\text{large}}(I_i, R_i)\right)}. \tag{5}$$

Noising data can be endowed with high score since the dual-model metric considers only the relative complexity with the neglect of generation quality. To address this issue, we utilize an LLM as referee for data sample scoring. This involves comparing each data sample in the same batch of diverse data samples generated by selected agent-pairs against a base data sample generated by base agent-pairs. There are three potential outcomes: the base data sample is better, the diverse data sample is better, or a tie, as shown in Appendix A.1. These outcomes are quantitatively assigned as quality scores, thereby avoiding collecting noising instruction samples:

$$\pi_{\text{llm}} = \begin{cases} 0, & \text{if the base data sample is better,} \\ 1, & \text{if the generated data sample is better,} \\ 0.5, & \text{if tie.} \end{cases} \tag{6}$$

Finally, the evaluation scores from both the LLM and the dual-model evaluation are combined to compute a final composite score:

$$\pi = \pi_{\text{llm}} \cdot \pi_{\text{dual}}. \tag{7}$$

This score determines the overall quality and suitability of data for enhancing the model's capabilities. The highest scoring data sample is then selected into dataset $T$ and Instruction Memory Bank as detailed in Section 3.3, ensuring that the chosen dataset maximizes potential improvements in model performance.

### 3.3 Evolving Star Agents

As mentioned in Section 3.1, we use the joint probability of instruction agents and response agents to regulate the invocation of each agent-pair. Considering the abilities and specialities of each LLM vary, however, sampling each agent-pair with the same probability is not optimal. We hence use the score from Section 3.2 to dynamically evolve the sampling probability. Additionally, since the generation performance of agent-pairs is task-dependent, we also propose an Instruction Memory Bank to select the most suitable agent-pair for particular tasks.

**Agent-Pair Sampling Evolution.**    Section 3.2 has introduced the score $\pi$, which effectively estimates the quality of generated samples. During each iteration, if the generated samples are of high quality, we will increase the sampling probability of the selected agent-pair, which is updated as follows:

$$\tilde{p}_{jk} = p_{jk} + \beta \cdot \pi(I_i, R_i),$$
$$p_{jk} \leftarrow \frac{\tilde{p}_{jk}}{\sum_{j,k} \tilde{p}_{jk}}. \tag{8}$$

The updated sampling probability for the agent-pair of the $j$-th instruction agent and $k$-th response agent that successfully process the $i$-th data sample will be used in the next iteration, where $\beta$ denotes the evolution rate. This formula adjusts the sampling probabilities based on the effectiveness demonstrated by agent-pairs in generating relevant data. Iterative updates ensure that as the synthesis process advances, the probability of selecting more effective agent-pairs increases, while less effective pairs are gradually phased out.

**Instruction Memory Bank Evolution.**    We establish an Instruction Memory Bank storing high-quality instructions aiming to accelerate sampling and relate the evolution with task data. When processing a data sample $(I_i, R_i)$, we perform a query in the Instruction Memory Bank for $I_i$, retrieving the top $n$ closest matches according to embedding similarity. The associated agent-pairs, identified as highly proficient for tasks similar to $I_i$, are then sampled. We sample $l$ agent-pairs from this pool using normalized probabilities to generate diverse data. Moreover, to foster the creation of a diverse dataset, additional $M - l$ agent-pairs are sampled from the remaining pool using their respective probabilities to assist in data synthesis. As a result, $M$ new samples are generated and then feed for data assessment. Subsequently, the Instruction Memory Bank will continuously evolve by incorporating tailored high-quality data, which get high socres as introduced in Section 3.2.

## 4    Experiments

We conducted extensive experiments to evaluate the proposed Star-Agents framework. A wide range of LLMs, benchmark datasets were used in our experiments to guarantee the robustness of our evaluation.

### 4.1    Setups

**Datasets.**    In alignment with the WizardLM [44], we adopted the Supervised Fine-Tuning (SFT) dataset, designated as the Evol-Instruct dataset, which consists of 70,000 instruction-response pairs. The instructions in this dataset were refined using "In-Depth Evolving" and "In-Breadth Evolving" methods, which were tailored to enhance the base instructions by adding intricate details or expanding the overall scope, respectively. To guarantee the fidelity of the data, ChatGPT was also integrated as generator into the refinement process. The quality of the instruction data from the Evol-Instruct dataset has been validated as superior [44, 25]; hence, our research continues to leverage these refined instructions. Employing the Star-Agents framework, our study invokes multiple LLMs to generate diverse and high-quality responses for these instructions. For further enriching our comparative analysis, we employed the Alpaca dataset [32], comprising 52,000 instruction-following samples. This dataset, developed under the self-instruct paradigm, utilizes the ChatGPT[2] instead of text-davinci-003 for a fair comparison [44].

---

[2]https://chatgpt.com/

Table 1: Typical LLMs utilized in Star-Agents.

| Model Famliy | Model Size | Data Size | Method | Source |
|--------------|------------|-----------|--------|--------|
| Phi [11] | 2.7B | 1.4T | Pretrain | Microsoft |
| ChatGLM [51] | 6B | 1T+ | SFT & RLHF | Zhipu AI |
| Gemma [34] | 7B | 6T | SFT & RLHF | Google |
| Mistral [12] | 7B | - | SFT | Mistral |
| Qwen [1] | 14B | - | SFT & RLHF | Alibaba |
| ChatGPT | - | - | SFT & RLHF | OpenAI |

**Models.** In response to the growing need for cost-effective inference of LLMs at the edge, our study explores the capabilities of target models scaled at 1B and 7B parameters. The 1B models, specifically the Pythia-1B [2], were trained on roughly 300 billion tokens derived from the Pile dataset. The 7B models, represented by the Llama-2-7B [35], were trained on an extensive corpus of 2 trillion tokens.

During our experiments, we integrated as generator a diverse array of LLMs, as detailed in Table 1. Our hypothesis posits that models from different development teams possess unique capabilities, yielding rich responses to identical prompts due to the diversity in their training data and strategies. For instance, the Phi2 [11] employed 1.4T tokens of meticulously curated textbook-like data without undergoing Reinforcement Learning with Human Feedback (RLHF) while the Gemma [34] was trained on 6T tokens primarily sourced from English web documents, mathematical content, and code, with subsequent fine-tuning through SFT and RLHF. To ensure the diversity and quality of generated data, we assembled LLMs trained by different teams, widely regarded for their exceptional performance. In pursuit of fostering the generation of data across varying levels of difficulty, the utilized LLMs range from 2.7B to 14B parameters, including even larger models via API access. For a fair comparison with the Evol-Instruct dataset, the most capable model employed was the ChatGPT, which was also used for generating responses within the Evol-Instruct dataset. Notably, the ChatGPT was also served as evaluator to compute the comparison score $\pi_{\text{llm}}$.

**Benchmarks.** To rigorously evaluate the instruction-following capabilities of AI models, we utilized three widely used benchmarks: MT-bench, Vicuna-bench, and the WizardLM testset. Specifically, MT-bench and Vicuna-bench are designed to test the models' competencies in various complex cognitive tasks, including mathematics, reasoning, complex format handling, and writing through both multi-turn and single-turn dialogues. The WizardLM testset, conversely, extends the evaluation to encompass diverse fields such as technology, biology, and law. It also features varied difficulty levels to facilitate a more nuanced comparison of models' performance disparities. Following established protocols, we employed the Fast-Chat [54] to assess model performances, with GPT-4 acting as the judge model.

**Baselines.** For baseline comparisons, we employed the Pythia-1B and Llama-2-7B, both trained using the Evol-Instruct datasets. The Alpaca datasets were also referenced for comparative analysis, alongside IFD [20] and Random select as an additional comparsion for data selection methods.

**Implementation Details.** We fine-tuned our models (Pythia-1B and Llama-2-7B) over three epochs using the Adam optimizer, with an initial learning rate of $2 \times 10^{-5}$, a maximum token count of 2048, and a batch size of 64. For the Star-Agents, 10 agent-pairs were employed.

## 4.2 Main Results

**GPT-4 Automatic Evaluation** Based on the findings summarized in Table 2, comprehensive training sessions were conducted for the Pythia-1B and Llama-2-7B models utilizing three distinct datasets: Alpaca, Evol-Instruct, and the optimally refined *Star Instruct* datasets. The latter was developed through the application of Star-Agents, which are derivatives of the Evol-Instruct datasets. Through comparative analyses with other contemporary state-of-the-art models, we observe that the SFT-aligned models employing the *Star Instruct* datasets consistently outperform nearly all aligned counterparts, across all evaluated model families.

Table 2: Results of different models on Vicuna-bench, WizardLM testset and MT-Bench.

| Model | Vicuna-Bench | WizardLM testset | MT-Bench | Average |
|---|---|---|---|---|
| **1B Models** | | | | |
| Pythia-1B [2] | 1.68 | 1.34 | 1.17 | 1.40 |
| OPT-1.3B [52] | 2.49 | 1.64 | 1.12 | 1.75 |
| Sheared-LLaMA-1.3B [42] | 2.73 | 1.86 | 1.59 | 2.06 |
| Pythia-1B-alpaca | 4.14 | 2.97 | 2.20 | 3.10 |
| Pythia-1B-evol_instruct | 5.07 | 3.55 | 2.56 | 3.73 |
| Pythia-1B-IFD [20] | 4.60 | 3.21 | 1.98 | 3.26 |
| Pythia-1B-Random | 5.13 | 3.39 | 2.35 | 3.62 |
| Pythia-1B-star_instruct | **5.93** | **3.90** | **2.69** | **4.17** |
| **7B Models** | | | | |
| Llama-2-7B [35] | - | - | 3.95 | - |
| zephyr-beta-sft [22] | - | - | 5.32 | - |
| mpt-7B-chat [22] | - | - | 5.45 | - |
| XGen-7B-8k-Inst [26] | - | - | 5.55 | - |
| sRecycled-Wiz-7B-v2 [17] | - | - | 5.56 | - |
| Llama-2-7B-alpaca | 6.33 | 5.08 | 3.63 | 5.01 |
| Llama-2-7B-evol_instruct | 7.27 | 6.57 | 5.21 | 6.35 |
| Llama-2-7B-star_instruct | **8.24** | **6.87** | **5.74** | **6.95** |

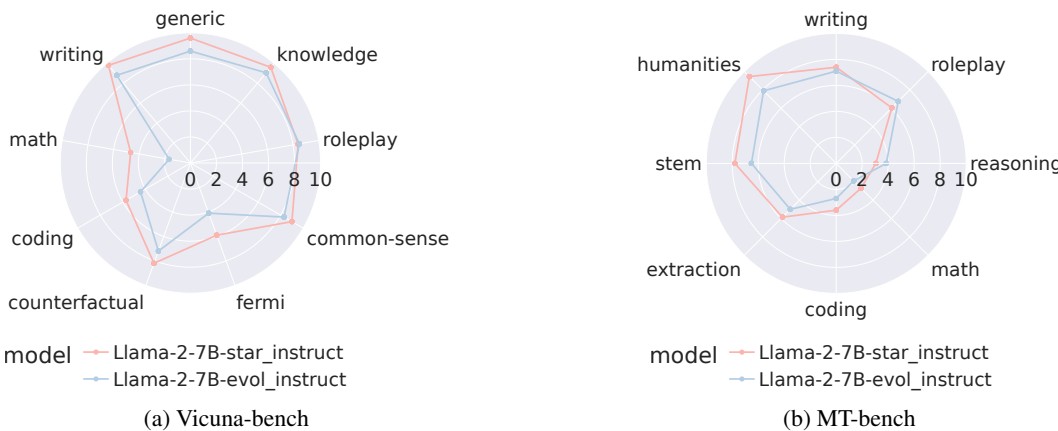

(a) Vicuna-bench        (b) MT-bench

Figure 4: Radar plot of detailed scores for Llama-2-7B-star_instrcut against the major baseline on different subtasks of (a) Vicuna-Bench and (b) MT-Bench.

Notably, at the 1B scale, models trained with the *Star Instruct* dataset demonstrate significant superiority, surpassing baselines across diverse evaluation datasets. Remarkably, in comparison to models trained with the Evol-Instruct dataset, those utilizing *Star Instruct* achieve an average absolute improvement of approximately 0.45, which is corresponding to a performance enhancement of about 12%. Additionally, when compared to models trained with the Alpaca dataset, our framework achieves an absolute improvement of 1 point, thereby affirming that the *Star Instruct* dataset is particularly well-suited for the Pythia-1B model, significantly boosting its operational efficacy. Additionally, within the 7b model category, the Llama-2-7B-star_instruct outperforms the sRecycled-Wiz-7B-v2 [17], which is trained on the Evol-Instruct dataset enhanced by Selective Reflection-Tuning. Figure 4a illustrates the Llama-2-7B-star_instruct's performance enhancements across nine metrics, with notable substantial improvements in math, coding and fermi problem-solving, where improvements surge up to 40%. A similar phenomenon can be observed in Figure 4b. Additionally, comparative examples of single-turn and multi-turn dialogues are provided in Appendix A.2, and the performance on the Open LLM Leaderboards of LLMs can be found in Appendix A.4.

Table 3: Impact of different components.

| Components | | | Average Score |
|---|---|---|---|
| Diversity | Data selection | Evolutiuon | |
| ✓ | ✓ | ✓ | **4.17** |
| ✓ | ✓ | ✗ | 3.97 |
| ✓ | ✗ | ✗ | 3.62 |
| ✗ | ✗ | ✗ | 3.73 |

Table 4: Imapct of the selection method.

| Model | Average Score |
|---|---|
| Pythia-1B-evol_instruct | 3.73 |
| Pythia-1B-IFD [20] | 3.26 |
| Pythia-1B-Random | 3.62 |
| Pythia-1B-star_instruct | **4.17** |

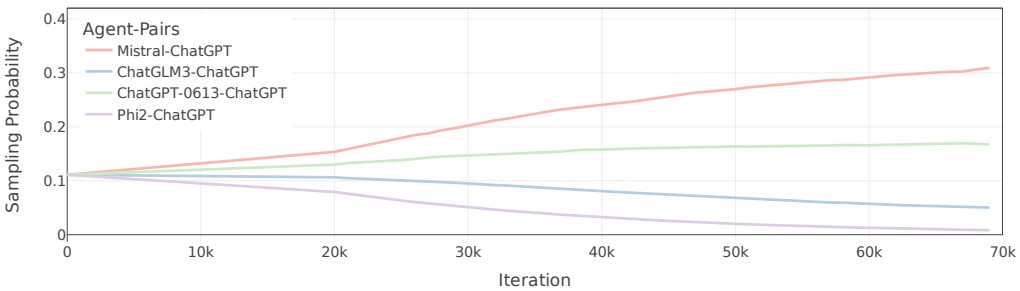

Figure 5: Evolution of the typical Agent-Pairs.

## 4.3 Ablation Study

**Main Components.** As illustrated in Table 3, we conducted ablation experiments on the three principal components within the Star-Agents framework. Results indicate that models using solely diversified datasets with random sampling yield a bit lower performance than the baseline. This occurs because the baseline employs data generated by ChatGPT, which is of high quality. In contrast, the diversified datasets draw from a variety of sources, making it challenging to ensure uniformly high quality. Thus, random sampling may introduce low-quality data, leading to diminished model performance. The inclusion of a data selection module subsequently leads to a recovery in model performance, suggesting that this module effectively selects high-quality data suitable for the model. Integration of the evolution strategy also provides a significant improvement, demonstrating that the evolution module can effectively select the most appropriate data generation agent-pairs from a complex array of candidate agent-pairs.

**Selection Method.** As demonstrated in Table 4, we evaluated a range of conventional selection methods, including both random selection and strategies informed by the IFD [20]. Our dual-model selection strategy significantly outperforms these approaches. Compared to random selection, our method achieves a significant improvement, registering an improvement exceeding 0.5 points on average across a variety of test sets. When compared with the IFD approach, our enhancement approaches a 0.9 point. These findings robustly validate the effectiveness of our dual-model selection strategy, illustrating its superior performance in refining model selection precision using diverse evaluation metrics.

**Evolution.** As depicted in Figure 5, we analyzed the sampling probability curves of typical agent-pairs throughout an iterative evolutionary process. Initially, each agent-pair began with a sampling probability of approximately 10%. Due to its robust performance, the Mistral-ChatGPT receives consistent rewards, which leads to a gradual increase in its sampling probability. By the completion of about 70,000 iterations, this probability has escalated to 30%. In stark contrast, the Phi2-ChatGPT undergoes a steady decline over the same period, with its sampling probability ultimately plummeting to near zero as it is progressively phased out. Concurrently, the ChatGLM3-ChatGPT exhibits a relatively stable trajectory, albeit with a slight downward trend. Evolutionary trajectories present significant discrepancy indicating different generation suitability of different generators on different tasks, where all the differences are captured by our evolution mechanism.

# 5 Conclusion

In this paper, we have presented the Star-Agents framework, an automated system for optimizing data to be optimally challenging for target LLMs. This framework has been applied to the open-source SFT datasets, and we conduct training sessions on a variety of model families, adjusting the data to enhance its efficacy. Our empirical investigations include a series of instruction tuning experiments that utilize both multiple baselines and specially optimized datasets on well-known models such as Pythia and LLaMA. Extensive experiments confirm the substantial impact of our method: the optimized tailored datasets result in an average performance enhancement of approximately 12%, with certain metrics, especially those involved in Fermi problem tasks exhibiting increases exceeding 40%, as substantiated by results on benchmarks such as MT-bench, Vicuna bench, and the WizardLM testset. These findings underscore the premise that strategically diverse and tailored data can profoundly improve model alignment and performance. In conclusion, our research details a highly effective automated framework that significantly augments dataset functionality, thus fostering more efficient model alignment.

**Limitations.** Our approach achieves remarkable performance improvements on single-turn instruction datasets. However, it has not yet been evaluated on multi-turn conversations. We hence leave the evaluation on multi-turn instruction datasets and validation on datasets with domain-specific instructions to our future work.

# Acknowledgements

The present research was partially supported by the National Key Research and Development Program of China (Grant No. 2023YFE0116400). We would like to thank the anonymous reviewers for their insightful comments.

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

# A Appendix

## A.1 Prompt Examples

Following the Fast-Chat [54], the prompts used in the data selection process are as listed in Table 5.

Table 5: Prompts of data selection for LLMs.

**System Prompt**: Please act as an impartial judge and evaluate the quality of the responses provided by three AI assistants to the user question displayed below. You should choose the assistant that follows the user's instructions and answers the user's question best. Your evaluation should consider factors such as the helpfulness, relevance, accuracy, depth, creativity, and level of detail of their responses. Begin your evaluation by comparing the two responses and provide a short explanation. Avoid any position biases and ensure that the order in which the responses were presented does not influence your decision. Do not allow the length of the responses to influence your evaluation. Do not favor certain names of the assistants. Be as objective as possible. After providing your explanation, output your final verdict by strictly following this format: "[[A]]" if assistant A is the bset, "[[B]]" if assistant B is the bset, and "[[C]]" for a tie.

**Prompt Template**:
[User Question]
<question>
[The Start of Assistant A's Answer]
<answer_a>
[The End of Assistant A's Answer]
[The Start of Assistant B's Answer]
<answer_b>
[The End of Assistant B's Answer]
[Final Verdict]:

## A.2 Case Study

Examples of the single-turn dialogue and multi-turn dialogue are presented in Table 6 and Table 7.

Table 6: Single-turn dialogue example.

**Instruction**: A tech startup invests $8000 in software development in the first year, and then invests half of that amount in software development in the second year. What's the total amount the startup invested in software development over the two years?

| **Llama-2-7B-evol_instruct**: The total amount the startup invested in software development over the two years is $12,000. | **Llama-2-7B-star_instruct**: To calculate the total amount the startup invested in software development over the two years, we need to find the total investment in software development for each year and then add them together. In the first year, the startup invested $8000 in software development. In the second year, the startup invested half of that amount, which is $4000. So, the total investment in software development for the two years is: $8000 (first year) + $4000 (second year) = $12000 Therefore, the total amount the startup invested in software development over the two years is $12000. |
|---|---|

Table 7: Multi-turn dialogue examples.

**Instruction**: Act as a math teacher. I will provide some mathematical equations or concepts, and it will be your job to explain them in easy-to-understand terms. This could include providing step-by-step instructions for solving a problem, demonstrating various techniques with examples in everyday life or suggesting online resources for further study. My first request is "I need help understanding how probability works."

**Llama-2-7B-evol_instruct**: Sure, I'd be happy to help! Probability is the likelihood of something happening, and it's a fundamental concept in math and science. To explain it simply, probability is the chance of something happening out of all possible outcomes. For example, if you flip a coin, the probability of getting heads is 1 out of 2, or 50%. Probability can be used to make predictions and calculate risks, and it's important in fields like finance, statistics, and data analysis.

To calculate probability, you can use different formulas depending on the type of problem you're solving. One common method is to use a formula called the probability mass function, which tells you the likelihood of getting a certain outcome. Another method is to use the probability density function, which tells you the likelihood of getting a certain range of outcomes.

There are many resources available online to help you learn more about probability, including video tutorials, interactive simulations, and practice problems. Some popular websites for learning probability include Khan Academy, Mathway, and Coursera. I hope this helps!

**Llama-2-7B-star_instruct**: Sure, I'd be happy to help explain probability! Probability is a measure of how likely an event is to occur. It's often expressed as a number between 0 and 1, where 0 represents a very unlikely event and 1 represents a very likely event. To calculate probability, you need to know the total number of possible outcomes and the number of favorable outcomes for the event you're interested in. For example, if you're flipping a fair coin, there are two possible outcomes (heads or tails), and the probability of getting heads is 1/2 or 0.5, because there's only one way to get heads out of two possible outcomes. In everyday life, probability can be used to calculate the likelihood of certain events, such as the chance of rain on a particular day or the probability of getting a certain grade on a test. It's also used in decision-making situations, like calculating the risk of a particular investment or the likelihood of success in a business venture. There are many resources available online to help you learn more about probability, such as video tutorials, practice problems, and interactive simulations. Some popular websites include Khan Academy, Coursera, and edX, which offer free courses and resources on probability and other mathematical concepts.

**Instruction**: What are the differences between Riemannian geometry and euclidean geometry?

**Llama-2-7B-evol_instruct**: Riemannian geometry and Euclidean geometry are two different types of geometries. In Euclidean geometry, the distance between two points is always a straight line, and the angles of a triangle always add up to 180 degrees. In Riemannian geometry, the distance between two points is not always a straight line, and the angles of a triangle can add up to more than 180 degrees.

Another key difference between the two geometries is the way they handle curved surfaces. In Euclidean geometry, a flat surface is a rectangle with parallel sides, while in Riemannian geometry, a surface can be curved in any way. This means that Riemannian geometry is better suited for describing the geometry of the universe, as it can handle the curvature of space-time.

Overall, Riemannian geometry is a more complex and advanced form of geometry, and it has many practical applications in fields such as physics and engineering.

**Llama-2-7B-star_instruct**: Great question! Riemannian geometry and Euclidean geometry are two different types of geometry that are used to describe and analyze the properties of spaces. The main differences between the two are as follows: 1. Metric: In Euclidean geometry, the distance between two points is defined by the Pythagorean theorem, which states that the sum of the squares of the two sides of a right triangle is equal to the square of the length of the hypotenuse. In Riemannian geometry, the distance between two points is defined by a metric, which is a function that describes how much the shortest path between two points is curved. 2. Curvature: Euclidean geometry deals with flat spaces where the curvature of the Earth's surface is negligible. In contrast, Riemannian geometry deals with curved spaces, such as the surface of the Earth, where the curvature plays a significant role in determining the properties of the space. 3. Geodesics: In Euclidean geometry, the shortest path between two points is a straight line. In Riemannian geometry, the shortest path between two points is called a geodesic, which can be curved and depends on the metric. 4. Tensor Fields: Riemannian geometry involves the use of tensor fields, which are mathematical objects that describe the geometry of a space. These fields are used to calculate the curvature of the space and the behavior of objects moving through it. Overall, Riemannian geometry and Euclidean geometry are two different approaches to understanding the properties of spaces, with Riemannian geometry being more applicable to curved spaces and Euclidean geometry being more applicable to flat spaces.

## A.3 Data Examples

Data examples are shown in Table 8.

Table 8: Diverse Data Examples.

| **Instruction**: Given some binary classification data, identify a suitable machine learning model that can be applied to this task. |
|---|

**Evol Instruct**: The suitable machine learning models for binary classification tasks include:
1) Logistic regression: It is a statistical method that uses a logistic function to model a binary dependent variable. It is a relatively simple and interpretable model that works well with linearly separable datasets.
2) Support Vector Machines (SVMs): It is a popular classifier that is used for binary classification tasks. SVMs can handle high-dimensional data and work well with datasets that are not linearly separable.
3) Random Forests: It is an ensemble learning method that works by constructing a multitude of decision trees at training time and outputs the class that is the mode of the classes output by the individual trees.
4) Neural networks: It is a type of machine learning model that is widely used for binary classification tasks. Neural networks can handle non-linear relationships between variables and are capable of learning complex patterns in the data.
The choice of the model depends on the particular characteristics of the data and the specific requirements of the project."

**Agent-Pair A**: There are several machine learning models that can be applied to binary classification tasks, depending on the nature of the data and the problem at hand. Here are a few popular ones:
1. Logistic Regression: This is a simple and effective model that can be used for binary classification problems. It works by estimating the probability of the positive class and using it to make predictions.
2. Decision Trees: Decision Trees are a popular machine learning model that can handle both categorical and numerical features. They work by recursively splitting the data into subsets based on the values of the input features.
3. Random Forest: Random Forest is an ensemble learning method that combines multiple decision trees to improve the accuracy of the model. It works by creating a set of decision trees and then averaging their predictions.
4. Support Vector Machines (SVM): SVM is a powerful model that can handle high-dimensional data. It works by finding the hyperplane that maximally separates the classes in the feature space.
5. Neural Networks: Neural Networks are a set of algorithms that can learn and classify from data. They can handle complex and nonlinear relationships between the input features and the output labels.
1The choice of the model depends on various factors such as the size and complexity of the dataset, the type of features, the computing resources, and the desired level of accuracy.

**Agent-Pair B**: For binary classification tasks, several machine learning models are suitable, depending on the size of the dataset, complexity of the problem, and the availability of features. Here are some popular models that can be applied:
1. Logistic Regression: A simple and interpretable model that works well for small to medium-sized datasets. It's linear and can handle both categorical and continuous features.
2. Decision Trees and Random Forest: These are easy to interpret and can handle both numerical and categorical data. Random Forest is an ensemble method that combines multiple decision trees, reducing overfitting.
3. Support Vector Machines (SVM): Effective for high-dimensional data and can handle non-linear relationships through kernel functions. SVMs are robust to outliers and can be used for both linear and non-linear classification.
4. Naive Bayes: A probabilistic model that assumes features are independent. It's fast, easy to implement, and works well for text classification or when features are not highly correlated.
5. K-Nearest Neighbors (KNN): A simple model that classifies based on the majority class of its nearest neighbors. It's non-parametric and can be useful for small datasets.
6. Neural Networks (e.g., Feedforward, Convolutional, or Recurrent Neural Networks): Deep learning models that can capture complex patterns in large datasets. They are powerful but require more data and computational resources.
7. Gradient Boosting Machines (GBM) or XGBoost: Ensemble methods that iteratively build weak learners (e.g., decision trees) to create a strong classifier. They are often used for structured data and can handle high dimensionality.
8. AdaBoost: Another boosting algorithm that combines weak learners to create a strong classifier. It's particularly useful when dealing with imbalanced datasets.
9. Neural Network Ensembles (e.g., stacking, blending): Combining predictions from multiple models can improve performance and robustness.
To choose the best model, it's essential to perform exploratory data analysis, split the data into training and testing sets, and evaluate the models using appropriate metrics (e.g., accuracy, precision, recall, F1-score, or AUC-ROC for imbalanced datasets). Cross-validation can help in selecting the optimal model and hyperparameters

**Agent-Pair C**: A suitable machine learning model for this task would be a Random Forest Classifier. This model is effective in handling both categorical and numerical data, and can handle large datasets with high dimensionality.

**Agent-Pair D**: One suitable machine learning model for binary classification tasks is the Logistic Regression model. Logistic Regression is commonly used for binary classification as it provides the probability of an instance belonging to a particular class. It is a simple and efficient algorithm that works well with linearly separable data.
Other popular machine learning models for binary classification tasks include Support Vector Machines (SVM), Decision Trees, Random Forest, Gradient Boosting Machines (GBM), and Neural Networks. The choice of the model depends on the complexity of the data, size of the dataset, interpretability requirements, and computational resources available.

## A.4 Performance on Open LLM Leaderboards

The performance on Open LLM Leaderboards is shown in Table 9.

Table 9: Performance on Open LLM Leaderboards.

| Model | ARC | HellaSwag | MMLU | TruthfulQA | Average |
|---|---|---|---|---|---|
| Wizardlm | 51.60 | 77.74 | 42.74 | 45.75 | 54.18 |
| Llama-2-7B-evol_instruct | 51.88 | 76.70 | 45.76 | 46.10 | 55.11 |
| Llama-2-7B-star_instruct | 54.44 | 77.64 | 46.94 | 46.13 | 56.29 |

## A.5 Computational Cost.

The computational overhead of our proposed method primarily depends on the inference computational load of the various LLMs used:

- Qwen-14B: During inference with a sequence length of 256 tokens, the computational load is approximately $4 \times 10^{12}$ Multiply-Add cumulations (MACs).

- Phi-2-2.7B: For the same sequence length, the inference computational load is around $7 \times 10^{11}$ MACs.

- ChatGPT: Given that ChatGPT is a proprietary model, we don't have details on its computational requirements.

Nonetheless, for estimation purpose, we can approximate the overall computational cost. Assuming an iterative process involving multiple LLMs (e.g., 10 LLMs) and a large dataset (e.g., 70,000 samples), the total computation without using our framework can be roughly estimated as:

- $4 \times 10^{12}$ FLOPs (Qwen-14B) $\times$ 10 LLMs $\times$ 70,000 samples = $2.8 \times 10^{18}$ MACs

While, when the Agent-Pairs Sampling and Instruction Memory Bank are employed, 5 of 10 LLMs are used to generate data , therefore, total computation can be significantly reduced and roughly estimated as:

- $4 \times 10^{12}$ FLOPs (Qwen-14B) $\times$ 5 LLMs $\times$ 70,000 samples = $1.4 \times 10^{18}$ MACs

