# OpenReview forum: "Star-Agents: Automatic Data Optimization with LLM Agents for Instruction Tuning"
_NeurIPS.cc/2024/Conference — NeurIPS 2024 poster_

### Official Review · Reviewer_4rks · 2024-07-10

**Soundness:** 3
**Presentation:** 3
**Contribution:** 3
**Rating:** 6
**Confidence:** 3

**Summary:**

The paper introduces the Star-Agents framework, an innovative system designed to enhance the quality of datasets used for instruction-tuning of large language models (LLMs). The framework addresses the challenges of collecting high-quality and diverse data by automating the process through multi-agent collaboration and assessment. The optimized datasets resulted in significant performance improvements. The research contributes to an advanced automated data optimization system that refines instruction samples with suitable complexity and diversity for LLMs, leading to more efficient model alignment and improved performance.

**Strengths:**

1. The paper presents a novel approach to enhancing the quality of datasets for instruction-tuning of LLMs. The concept of using multiple agents to generate diverse instruction data and the dual-model evaluation strategy for data selection are innovative.

2. The paper includes comprehensive experiments with different models, showcasing the effectiveness of the Star-Agents framework.

3. The reported performance improvements of up to 12% on average, and over 40% in specific metrics, highlight the practical significance of the research.

**Weaknesses:**

1. This paper utilizes multi-agent collaboration, but it seems that the paper does not explore the impact of different numbers of agents or agent pairs on the results. The authors should also specify the exact number of agents used in the experiments in the "Implementation Details" section.

**Questions:**

1. Could you provide some specific cases? For example, demonstrate how the outputs from different agent pairs differ.

2. What impact would removing some or adding additional agent pairs have on the results?

**Limitations:**

The authors have mentioned the limitations of this work in the paper and have also suggested potential research directions for addressing these limitations.

---

> ### Author Rebuttal · Authors · 2024-08-07
>
> Thanks for the constructive comments.
>
> **Q1**: Different numbers of agents or agent pairs on the results.
>
> **A1**:  We have conducted experiments to explore the impact of varying the number of agent pairs on the results. As shown in Table 1, we observed that as the number of agent pairs decreases, the model's performance exhibits a corresponding decline. This finding highlights the importance of the number of agents in achieving optimal results. We will include these experimental results and a discussion of their implications in the revised manuscript.
>
> Table 1. Performance using different agent pair
>
> | model                          | agent-pair | Vicuna-bench | Wizardlm-bench | MT-bench | Average |
> | ------------------------------ | ---------- | ------------ | -------------- | -------- | ------- |
> | Pythia-1B-evol_instruct        | 1          | 5.07         | 3.55           | 2.56     | 3.73    |
> | Pythia-1B-star_instruct_2_pair | 2          | 5.23         | 3.88           | 2.69     | 3.93    |
> | Pythia-1B-star_instruct_5_pair | 5          | 5.29         | 3.87           | 2.78     | 3.98    |
> | Pythia-1B-star_instruct_10_pair | 10         | 5.93         | 3.90           | 2.69     | 4.17    |
>
> **Q2**: The authors should also specify the exact number of agents used in the experiments in the "Implementation Details" section.
>
> **A2**: Thank you for your insightful suggestion. In our experiments, we utilized a total of 10 agent-pairs. We will include this specific information in the "Implementation Details" section of the revised manuscript to ensure clarity and completeness.
>
> **Q3**: Could you provide some specific cases? For example, demonstrate how the outputs from different agent pairs differ.
>
> **A3**: we have included a specific case in the table below to demonstrate how the outputs from different agent pairs differ briefly. Due to the character limitations of this rebuttal, more details are provided in the supplementary PDF attached to the global rebuttal. We kindly ask you to review the detailed examples there.
>
> - Instruction: Can you suggest a highly specialized conference dedicated to exploring advanced machine learning algorithms and their practical applications in the vast field of artificial intelligence? We are looking for keynote speakers who are prominent figures in the field of AI research, and the event should provide attendees with a chance to participate in immersive, hands-on workshops, as well as poster sessions highlighting groundbreaking research in this field. Additionally, the conference should offer insights into the latest trends and innovations in the world of machine learning.
> - Agent-Pair A: I recommend checking out the Conference on Neural Information Processing Systems (NeurIPS). It is one of the most prestigious conferences in the field of machine learning and artificial intelligence. NeurIPS features keynote speakers from top universities and research institutions, as well as hands-on workshops and poster sessions showcasing cutting-edge research in the field. The conference also offers a chance to learn about the latest trends and innovations in machine learning.
> - Agent-Pair B: The conference you are looking for is the International Conference on Machine Learning (ICML). It is a highly specialized conference that focuses on exploring advanced machine learning algorithms and their practical applications in the field of artificial intelligence. The conference is organized by the International Machine Learning Society (IMLS) .......
> - Agent-Pair C: I would highly recommend the NeurIPS (Conference on Neural Information Processing Systems) conference for your needs. NeurIPS is one of the largest and most prestigious annual conferences in machine learning and artificial intelligence. It attracts leading researchers, engineers, and students in the field, providing an ideal setting for sharing new research, ideas, and innovations..........
> - Agent-Pair D: Certainly! I recommend the \"Machine Learning and Artificial Intelligence Conference\" which will be held in San Francisco, California. The conference will feature keynote speakers such as Andrew Ng, Yoshua Bengio, and Geoffrey Hinton, who are all renowned experts in the field of AI research. Attendees will have the opportunity to participate in immersive, hands-on workshops, as well as poster sessions highlighting groundbreaking research in machine learning. The conference will also offer insights into the latest trends and innovations in the world of machine learning.
> - Agent-Pair E: One highly specialized conference that meets your criteria is the \"International Conference on Machine Learning\" (ICML). ICML is one of the premier events in the field of machine learning and attracts top researchers, practitioners, and industry leaders. The conference features keynote speeches by renowned experts, in-depth workshops, and numerous poster sessions showcasing cutting-edge research.\n\nHere are some key details about ICML:........Another conference that you might consider is the \"NeurIPS\" (Conference on Neural Information Processing Systems), which is another major event in the AI and machine learning community. NeurIPS has a similar format and focus on advanced research and practical applications.
> - Agent-Pair F: .........
>
> **Q4**: What impact would removing some or adding additional agent pairs have on the results?
>
> **A4**: Thank you very much for your insightful suggestion. We have conducted additional experiments to investigate the impact of varying the number of agent pairs on our model's performance. As illustrated in Table 1, we observed that reducing the number of agent pairs tends to result in a gradual decline in the model's effectiveness. This trend indicates that the performance of our model is positively correlated with the number of agent pairs. We hope these findings address your concerns satisfactorily.

---

> > ### Comment · Reviewer_4rks · 2024-08-12
> > **Response to the authors**
> >
> > Thanks to the authors for resolving my concern. For the Q1, from the table, it seems that 10 agent pairs is not the number that achieve the top of performance. What still concerns me is the balence of performance and cost (if more agent pairs cost more).

---

> > > ### Author Response · Authors · 2024-08-13
> > > **Thanks for your feedback!**
> > >
> > > Dear Reviewer 4rks,
> > >
> > > Thank you for your valuable feedback. We understand your concern regarding the balance between performance and cost when increasing the number of agent pairs.
> > >
> > > To address this, we conducted additional experiments. As shown in Table 2, we observed that the performance improvement starts to plateau when the number of agent pairs increases to 8. While the model's performance is highest with 10 agent pairs, the difference between 8 and 10 pairs is marginal. Considering the trade-off between performance and cost, we believe that using 8–10 agent pairs is a reasonable and effective choice. This range provides a good balance, ensuring high performance while keeping the cost manageable.
> > >
> > > Table 2. Performance using different agent pairs
> > > | model                           | agent-pair | Vicuna-bench | Wizardlm-bench | MT-bench | Average |
> > > | ------------------------------- | ---------- | ------------ | -------------- | -------- | ------- |
> > > | Pythia-1B-evol_instruct         | 1          | 5.07         | 3.55           | 2.56     | 3.73    |
> > > | Pythia-1B-star_instruct_2_pair  | 2          | 5.23         | 3.88           | 2.69     | 3.93    |
> > > | Pythia-1B-star_instruct_5_pair  | 5          | 5.29         | 3.87           | 2.78     | 3.98    |
> > > | Pythia-1B-star_instruct_7_pair  | 7          | 5.48         | 3.92           | 2.79     | 4.06    |
> > > | Pythia-1B-star_instruct_8_pair  | 8          | 5.69         | 3.92           | 2.78     | 4.13    |
> > > | Pythia-1B-star_instruct_9_pair  | 9          | 5.80          | 3.91           | 2.77     | 4.16    |
> > > | Pythia-1B-star_instruct_10_pair | 10         | 5.93         | 3.9            | 2.69     | 4.17    |

---

> ### Author Response · Authors · 2024-08-11
> **Thanks for the comments.**
>
> Dear Reviewer 4rks,
>
> We sincerely appreciate the time you’ve taken to provide valuable feedback on our paper. In our rebuttal, we have thoroughly addressed all of your initial concerns and included the requested experimental results. If you have any further questions or concerns, we would be happy to discuss them with you. Additionally, we welcome any new suggestions or comments you may have!

---

### Official Review · Reviewer_b1DV · 2024-07-12

**Soundness:** 3
**Presentation:** 3
**Contribution:** 3
**Rating:** 6
**Confidence:** 4

**Summary:**

The paper introduces the "Star-Agents" framework, designed to optimize data for instruction tuning in large language models (LLMs). This system automates the process of enhancing dataset quality by employing a multi-agent approach. Empirical studies demonstrate that the optimized datasets lead to significant performance improvements in models such as Pythia and LLaMA.

**Strengths:**

1. The motivation is clear and it is easy to follow their method.
2. The framework's effectiveness is validated through extensive experiments, showing significant performance improvements in various benchmarks.

**Weaknesses:**

1. What is the overhead of this proposed method, like wall-clock time?
2. stability: can this method be scalable to large scale dataset optimization, like web-crawl data?

**Questions:**

Please refer to the above.

**Limitations:**

Yes

---

> ### Author Rebuttal · Authors · 2024-08-07
>
> Thanks for the constructive comments.
>
> **Q1**：What is the overhead of this proposed method, like wall-clock time?
>
> **A1**：Thank you for your insightful question regarding the overhead of the proposed method.
>
> The computational overhead of our proposed method primarily depends on the inference computational load of the various  Large Language Models (LLMs) used. To provide a clearer understanding, let us consider specific LLMs use in this paper:
>
>  - LLM Qwen-14B: During inference with a sequence length of 256 tokens, the computational load is approximately 4×10^12 floating point operations (FLOPs).
>  - LLM Phi-2-2.7B: For the same sequence length, the inference computational load is around 7×10^11 FLOPs.
>  - LLM ChatGPT: Given that ChatGPT is a proprietary model, we lack precise data on its computational requirements.
>
> Nonetheless, for estimation purposes, we can approximate the overall computational cost. Assuming an iterative process involving multiple LLMs (e.g.,  10 LLMs) and a large dataset (e.g., 70,000 samples), the total computation without our methods can be roughly estimated as:
>
> 4×10^12 FLOPs (Qwen-14B) × 10  LLMs  × 70,000 samples = **2.8×10^18 FLOPs**
>
> While, when the Agent-Pairs Sampling and Instruction Memory Bank were employed,  5 of 10 LLMs are employed to generate data ,  therefore, total computation can be significant reduced and roughly estimated as:
>
> 4×10^12 FLOPs (Qwen-14B) × **5**  LLMs  × 70,000 samples = **1.4×10^18 FLOPs**
>
> This estimation highlights the significant computational requirements, largely dominated by the inference processes of the LLMs. Other components of our method contribute negligible computational overhead compared to the inference load of the LLMs.
>
> **Q2**：stability: can this method be scalable to large scale dataset optimization, like web-crawl data?
>
> **A2**：We appreciate your inquiry about the scalability of our method. Our approach focuses on enhancing the quality and diversity of instruction datasets, and it is inherently scalable to larger datasets. The methodology we proposed is not constrained by dataset size, thus it can be effectively applied to extensive datasets.
>
> Specifically, there are existing studies that leverage large models to synthesize and process pre-training data [r1]. Our method can be integrated with these existing approaches to optimize and handle large-scale pre-training datasets such as web-crawl data. This integration is a promising area for our future research, and we are committed to exploring this direction further.
>
> References:
>
> [r1] : Nemotron-4 340B Technical Report[J]. arXiv preprint arXiv:2406.11704, 2024.

---

> > ### Comment · Reviewer_b1DV · 2024-08-13
> >
> > Thanks for your response and it would be great if we can add this in the camera ready version. I will keep my score.

---

> > > ### Author Response · Authors · 2024-08-14
> > > **Thanks for your feedback!**
> > >
> > > Dear Reviewer b1DV,
> > >
> > > Thanks for your feedback and valuable suggestions!  We will ensure that the recommended additions are included in the camera-ready version of the paper.
> > >
> > > Regards

---

> ### Author Response · Authors · 2024-08-11
> **Thanks for the comments.**
>
> Dear Reviewer b1DV,
>
> We sincerely appreciate the time you took to provide valuable comments on our paper. In our rebuttal, we have addressed all of your initial concerns and included the requested experimental results. If you still have any question or concern, we would be glad to discuss them with you. Additionally, we welcome any new suggestions or comments you may have!
>
> Regards

---

### Official Review · Reviewer_kHqZ · 2024-07-15

**Soundness:** 3
**Presentation:** 4
**Contribution:** 3
**Rating:** 6
**Confidence:** 3

**Summary:**

The Star-Agents framework presents an advanced approach for enhancing data quality in instruction-tuning of large language models (LLMs) through multi-agent collaboration and automated assessment. By generating diverse instruction data using various LLM agents and evaluating it with a Dual-model metric, this framework ensures both diversity and quality. It dynamically refines its processes to prioritize more effective agents, leading to significant performance improvements. Experimental results show an average increase of 12% in model performance, with some metrics seeing increases up to 40%. This method addresses common issues in data generation such as lack of stylistic variety and overly complex datasets, making it a robust solution for optimizing LLM training datasets.

**Strengths:**

The paper stands out for its innovative dual-model scoring system, which refines data quality assessment for large language models (LLMs) and is a key contribution to the field. It demonstrates state-of-the-art results, with the framework achieving an average performance increase of 12% and up to 40% in specific metrics, highlighting its effectiveness. Additionally, the paper is clearly written and well-structured, making complex concepts accessible and easy to follow, which enhances its academic impact and usability.

**Weaknesses:**

I could not find any weakness apart from mentioned in the questions below.

**Questions:**

What is the size and composition of the datasets used in your experiments?
Have you compared your framework's performance using standard benchmarks like MMLU or those from Hugging Face?
What would be the impact of using only Mistral-ChatGPT for 70,000 iterations on the diversity and quality of the generated data?

**Limitations:**

The authors acknowledge a limitation in their framework, noting that it is currently designed for optimizing single-turn interactions. They suggest that extending this approach to multi-turn scenarios could further enhance its applicability and effectiveness, addressing more complex dialogue systems where contextual continuity and conversational depth are critical.

---

> ### Author Rebuttal · Authors · 2024-08-07
>
> Thanks for your constructive comments.
>
> **Q1**：What is the size and composition of the datasets used in your experiments?
>
> **A1**：The datasets used in our experiments consist of 70,000 samples based alpaca evol instruct datasets [r1]. Each sample is paired with an instruction and a corresponding response.  Here is an example:
>
> - Instruction:  Can you suggest a highly specialized conference dedicated to exploring advanced machine learning algorithms and their practical applications in the vast field of artificial intelligence? We are looking for keynote speakers who are prominent figures in the field of AI research, and the event should provide attendees with a chance to participate in immersive, hands-on workshops, as well as poster sessions highlighting groundbreaking research in this field. Additionally, the conference should offer insights into the latest trends and innovations in the world of machine learning.
>
> - Response:  I recommend checking out the Conference on Neural Information Processing Systems (NeurIPS). It is one of the most prestigious conferences in the field of machine learning and artificial intelligence. NeurIPS features keynote speakers from top universities and research institutions, as well as hands-on workshops and poster sessions showcasing cutting-edge research in the field. The conference also offers a chance to learn about the latest trends and innovations in machine learning.
>
> **Q2**: Have you compared your framework's performance using standard benchmarks like MMLU or those from Hugging Face?
>
> **A2**: We evaluated our framework on the standard benchmarks from the Open LLM Leaderboard (including MMLU). As shown in Table 1, although the training datasets we used in our paper focus more on conversational performance, our framework still achieved improvements across multiple standard test sets.
>
> Table 1. Performance on Open LLM Leaderboard
>
> | Model                    | ARC   | HellaSwag | MMLU  | TruthfulQA | Average |
> | ------------------------ | ----- | --------- | ----- | ---------- | ------- |
> | Wizardlm                 | 51.60 | 77.7      | 42.7  | 44.7       | 54.18   |
> | Llama-2-7B-evol_instruct | 51.88 | 76.70     | 45.76 | 46.10      | 55.11   |
> | Llama-2-7B-star_instruct | 54.44 | 77.64     | 46.94 | 46.13      | 56.29 |
>
> **Q3**: What would be the impact of using only Mistral-ChatGPT for 70,000 iterations on the diversity and quality of the generated data?
>
> **A3**: Thank you for your insightful question regarding the impact of using Mistral-ChatGPT for 70,000 iterations on the diversity and quality of the generated data. We have conducted a detailed analysis to address your concerns.
>
> Firstly, we generated 70,000 data samples exclusively using the agent-pair Mistral-ChatGPT. In terms of diversity, we observed a decline in the variation of responses for the same instructions. Regarding quality, some of the data generated was of lower quality compared to using the all of agent-pairs, but it was still superior to the baseline data.
>
> To further assess the implications, we trained the model using this generated dataset. The performance metrics are presented in the Table 2. The results demonstrate that relying on Mistral-ChatGPT for data generation negatively impacts the model's overall capability compared to using the full set of agent-pairs, but it is still superior to the baseline. This confirms the effectiveness of our approach.
>
> Table 2. Performance of Mistral-ChatGPT agent pair
> | Model                   | Vicuna-bench | Wizardlm-testset | MT-bench | Average |
> | ----------------------- | ------------ | ---------------- | -------- | ------- |
> | Pythia-1B-evol_instruct | 5.07         | 3.55           | 2.56     | 3.73    |
> | Pythia-1B-star_mistral  | 5.23         | 3.88             | 2.69     | 3.93    |
> | Pythia-1B-star_instruct | 5.93         | 3.90             | 2.69     | 4.17    |
>
>
> References:
>
> [r1] WizardLM: Empowering large pre-trained language models to follow complex instructions[C]//The Twelfth International Conference on Learning Representations. 2024.

---

> > ### Comment · Reviewer_kHqZ · 2024-08-09
> >
> > Thank you for addressing most of my concerns. I have few comments.
> > "Thank you for your insightful question regarding the impact of using Mistral-ChatGPT for 70,000 iterations on the diversity and quality of the generated data. We have conducted a detailed analysis to address your concerns.
> >
> > Firstly, we generated 70,000 data samples exclusively using the agent-pair Mistral-ChatGPT. In terms of diversity, we observed a decline in the variation of responses for the same instructions. Regarding quality, some of the data generated was of lower quality compared to using the all of agent-pairs, but it was still superior to the baseline data.
> >
> > To further assess the implications, we trained the model using this generated dataset. The performance metrics are presented in the Table 2. The results demonstrate that relying on Mistral-ChatGPT for data generation negatively impacts the model's overall capability compared to using the full set of agent-pairs, but it is still superior to the baseline. This confirms the effectiveness of our approach."
> >
> >
> > Here what metric did you use to calculate diversity?

---

> ### Author Response · Authors · 2024-08-11
> **Thanks for your feedback!**
>
> Dear Reviewer kHqZ,
>
> Thank you for your thoughtful feedback and valuable comments. We appreciate the opportunity to further clarify our methodology.
>
> Regarding the metric used to calculate diversity, we employed cosine similarity as the primary metric. Specifically, we calculated the cosine similarity between the sentence embeddings of the datasets. The diversity metric is inversely related to the cosine similarity; hence, lower cosine similarity indicates higher diversity in the generated data.
> In our analysis, we found that the Mistral-ChatGPT dataset exhibited a cosine similarity of approximately 20.8%, which indeed suggests a higher degree of similarity (and thus lower diversity) when compared to the dataset generated using all agent pairs, which had a cosine similarity of 6.9%.
>
> We hope this clarification adequately addresses your concerns, and we are grateful for your guidance in improving the clarity and rigor of our work.

---

### Official Review · Reviewer_piXU · 2024-07-17

**Soundness:** 2
**Presentation:** 2
**Contribution:** 2
**Rating:** 5
**Confidence:** 4

**Summary:**

The paper presents a framework for enhancing the quality of instruction datasets used for tuning large language models (LLMs). The proposed framework, Star-Agents, leverages multi-agent collaboration to generate, evaluate, and refine instruction data automatically. The approach comprises three main components:

1. **Diverse Data Generation**: The framework employs multiple LLM agents to generate varied instruction data. Each agent pair, consisting of distinct instruction and response agents, creates diverse data samples to enrich the dataset.
2. **Dual-model Evaluation**: This component introduces a two-tiered evaluation mechanism using both small and large models. The evaluation metric considers the difficulty and quality of the generated data, ensuring it is challenging yet manageable for the target model.
3. **Dynamic Refinement**: The framework dynamically adjusts the sampling probabilities of agent pairs based on their performance, promoting the selection of the most effective agents for data generation.

Empirical studies demonstrate the optimized datasets generated by Star-Agents led to performance improvements, with an average increase of 12% and up to 40% in specific benchmarks such as MT-bench, Vicuna-bench, and the WizardLM testset.

------------------------------------------------------------------------------------------------------------------------------------------------------------
Thank you for your reply and I have updated my score accordingly.

**Strengths:**

**Originality**
The paper introduces a framework, Star-Agents, for automatic data optimization in instruction tuning for large language models (LLMs). This originality arises from:

- **Multi-Agent Collaboration**: Utilizing multiple LLM agents to generate diverse and high-quality instruction data. This approach addresses the common limitations of single-model data generation methods, ensuring a richer and more varied dataset.
- **Dual-Model Evaluation**: Implementing a dual-model evaluation strategy that assesses both the difficulty and quality of the generated data. This innovative metric ensures the data is challenging yet manageable, enhancing the instruction tuning process.
- **Dynamic Refinement**: The dynamic adjustment of sampling probabilities for agent pairs based on performance is a creative mechanism that optimizes the data generation process over time.

**Quality**
The paper is supported by empirical studies. The framework is tested using instruction-tuning experiments with two models, including Pythia and LLaMA. The results consistently show performance improvements, validating the quality of the proposed method.

**Clarity**
The paper is clearly written and well-structured, facilitating easy understanding of the proposed framework.

**Weaknesses:**

**Limited Dataset Evaluation**
The paper evaluates the Star-Agents framework on a limited set of datasets, which may not fully capture the framework's robustness and generalizability. Specifically:

- **Small Evaluation Datasets**: The datasets used for evaluations are relatively small. Evaluating the framework on larger, more diverse datasets would provide a better understanding of its effectiveness across different data scales and domains.

**Complexity of the Method**
The Star-Agents framework involves multiple stages and the use of several teacher models, which introduces complexity:

- **Multiple Teacher Models**: The requirement for multiple LLM agents increases the computational cost and complexity of the implementation. Simplifying the framework or exploring methods to reduce the number of required teacher models could enhance its practical applicability.
- **Three-Stage Process**: The three-pronged approach, while thorough, may be overly complex for some applications. Streamlining the process without sacrificing performance gains could make the framework more accessible and easier to implement.

**Lack of Human Evaluations**
The paper does not include human evaluations of test data, which is crucial for assessing the practical quality and usability of the model.

"Absence of Standard Benchmark Results"
The framework's performance is not evaluated on widely recognized benchmarks, limiting the comparability of its results:

- **Standard Benchmarks**: Including evaluations on standard benchmarks such as GSM8K, HumanEval, or other well-known datasets would provide a clearer comparison with existing methods. This would help situate the Star-Agents framework within the broader context of instruction tuning and LLM performance.

**Questions:**

What will be the results if base models are larger, e.g. llama 13b or llama 70b?

**Limitations:**

See more details in Weaknesses section.

---

> ### Author Rebuttal · Authors · 2024-08-07
>
> Thanks for your constructive comments.
>
> **Q1**: Small Evaluation Datasets
>
> **A1**: We followed several seminal works[r1,r2], and used widely accepted datasets such as Mt-bench, Vicuna-bench, and the WizardLM testset for our evaluations. These datasets are commonly utilized to assess the effectiveness of LLM's instruction tuning capabilities.
> To further substantiate the robustness and generalizability of our framework, we also conducted evaluations using datasets from the standard benchmark Open LLM Leaderboard. As shown in Table 1, despite the primary focus of our datasets on instruction tuning capabilities, our method demonstrated  improvements across multiple test sets. This indicates that the Star-Agents framework not only excels in instruction tuning tasks but also performs well on a broader range of datasets.
>
> Table 1. Performance on Open LLM Leaderboard
>
> |Model|ARC|HellaSwag|MMLU|TruthfulQA|Average|
> |-|-|-|-|-|-|
> |Wizardlm-7B|51.60|77.7|42.7|44.7|54.18|
> |Llama-2-7B-evol_instruct|51.88|76.70|45.76|46.10|55.11|
> |Llama-2-7B-star_instruct|54.44|77.64|46.94|46.13|56.29|
>
> **Q2**: Multiple Teacher Models
>
> **A2**:  Compared to a single LLM, using multiple LLMs may increase the time complexity. However, we employed a model parallelism approach during the data generation process, which does not significantly increase the latency of our method while effectively improving the results. Additionally, we have proposed a selection strategy to reduce complexity.
> We introduced a dynamic selection strategy for LLMs, where during each data generation process, we sample based on the past performance of the LLMs. The sampled LLMs are responsible for generating diverse data. By limiting the number of activated LLMs each time, we can ensure the quality of data generation while significantly reducing the resources required for LLMs inference. In the future, we will continue to research simplified methods to enhance the efficiency of our approach.
>
> **Q3**: Three-Stage Process
>
> **A3**: Thank you for your comments.
> Firstly, Our three-stage method ensures diverse, high-quality data by leveraging multiple LLMs for varied data generation, using dual-model evaluation to identify high-quality, tailored data, and employing an evolution strategy to adjust sampling probabilities for efficient, task-specific data generation. Each stage serves a distinct purpose, and has a clear and justified function, and their integration ensures a streamlined process in practice.
> We acknowledge the need to simplify the process. Our future work will focus on optimizing data generation, reducing computational costs, and exploring automated tools to enhance usability. These improvements aim to make the Star-Agents framework easier to implement without compromising performance.
>
> **Q4**: Lack of Human Evaluations
>
> **A4**: In our paper, we conducted evaluations on three different test sets using the LLM-as-a-judge method based on GPT-4, as outlined in [r1]. According to [r1], the LLM-as-a-judge approach based on GPT-4 achieves an agreement rate of 85% with human evaluations, surpassing the inter-rater agreement rate among humans, which is approximately 80%. Moreover, numerous seminal works, such as [r3], and [r4], have successfully employed GPT-4 in LLM-as-a-judge to replace human evaluations.
> Additionally, we conducted a manual inspection of the evaluation results and found that the trends observed with GPT-4 evaluations align with those of human evaluations. Therefore, we adopted the LLM-as-a-judge method based on GPT-4 for our evaluations in this paper.
>
> **Q5**：Standard Benchmarks:
>
> **A5**：In response to this suggestion, we have conducted additional evaluations of the Star-Agents framework using the datasets available on the standard benchmark Open LLM Leaderboard. As shown in Table 1, our framework demonstrated improvements across various test sets, even though our chosen datasets were focused on dialogue performance. We believe that these additional benchmark results will provide a clearer and more robust comparison, demonstrating the effectiveness and versatility of the Star-Agents framework in various scenarios.
>
> **Q6**：What will be the results if base models are larger, e.g. llama 13b or llama 70b?
>
> **A6**：Due to time and computational constraints, we conducted preliminary experiments on the LLaMA2-13B model,  as presented in Table 2 . Our method shows a significant improvement compared to the baseline. Additionally, in the MT-bench First-turn evaluation, Llama-2-13B-star_instruct achieved a score of 6.98, which is close to the Llama2-70B-Chat score of 6.99, a much larger model that has undergone SFT and DPO. While, the teacher models used in our experiments were mostly smaller than 13B. For larger models, it is essential to incorporate more powerful teacher models to further enhance their performance effectively.
> Table2. Performance of Llama2-13B
> |Model|MT-bench|
> |-|-|
> |Llama-2-13B-evol_instruct|5.88|
> |Llama-2-13B-star_instruct|6.28|
>
> References:
>
> [r1] Judging llm-as-a-judge with mt-bench and chatbot arena[J]. Advances in Neural Information Processing Systems, 2024, 36.
>
> [r2] WizardLM: Empowering large pre-trained language models to follow complex instructions[C]//The Twelfth International Conference on Learning Representations. 2024.
>
> [r3] Mixtral of experts[J]. arXiv preprint arXiv:2401.04088, 2024.
>
> [r4] Minicpm: Unveiling the potential of small language models with scalable training strategies[J]. arXiv preprint arXiv:2404.06395, 2024.

---

> ### Author Response · Authors · 2024-08-11
> **Thanks for the comments.**
>
> Dear Reviewer piXU,
>
> We sincerely appreciate the time and effort you invested in providing valuable feedback on our paper. In our rebuttal, we have thoroughly addressed all of your initial concerns and included the requested experimental results. If you have any further questions or concerns, we would be happy to discuss them with you. Additionally, we welcome any new suggestions or comments you might have!
>
> Regards

---

### Author Rebuttal · Authors · 2024-08-07

Dear Reviewers,

Thank you for the overall positive reviews and helpful feedback, which we have incorporated to improve our work. If any remaining doubts exist, we encourage the reviewers to engage in the discussion so we can clarify them. If all concerns have been resolved, we kindly ask the reviewers to consider raising their scores.

Best Regards,

Submission 15632 Authors

---

### Decision · Program_Chairs · 2024-09-25

**Decision:**

Accept (poster)

**Comment:**

The paper introduces Star-Agents, an innovative framework designed to optimize datasets for instruction-tuning large language models. The approach involves a multi-LLM process to generate diverse instruction data, a dual-model evaluation mechanism to assess difficulty and quality, and a refinement process which prioritizes more effective LLMs. The reviewers generally agree on the strengths of the paper, particularly the IFD-based data selection mechanism, with its impact confirmed in ablations. The approach shows substantial improvements consistently across multiple benchmarks. Despite minor concerns regarding computational cost and the complexity of the framework, this looks like a compelling approach to dataset optimization for LLM instruction-tuning.